# Cutaneous Melanoma: A Review of Multifactorial Pathogenesis, Immunohistochemistry, and Emerging Biomarkers for Early Detection and Management

**DOI:** 10.3390/ijms242115881

**Published:** 2023-11-01

**Authors:** Laura Maria Gosman, Dana-Antonia Țăpoi, Mariana Costache

**Affiliations:** 1Doctoral School, Carol Davila University of Medicine and Pharmacy, 020021 Bucharest, Romania; laura.gosman@drd.umfcd.ro; 2Department of Pathology, Saint Pantelimon Clinical Emergency Hospital, 021659 Bucharest, Romania; 3Department of Pathology, Carol Davila University of Medicine and Pharmacy, 020021 Bucharest, Romania; mariana.costache@umfcd.ro; 4Department of Pathology, University Emergency Hospital, 050098 Bucharest, Romania

**Keywords:** cutaneous melanoma, immunohistochemistry, genetic mutations, microRNA, exosomes

## Abstract

Cutaneous melanoma (CM) is an increasingly significant public health concern. Due to alarming mortality rates and escalating incidence, it is crucial to understand its etiology and identify emerging biomarkers for improved diagnosis and treatment strategies. This review aims to provide a comprehensive overview of the multifactorial etiology of CM, underscore the importance of early detection, discuss the molecular mechanisms behind melanoma development and progression, and shed light on the role of the potential biomarkers in diagnosis and treatment. The pathogenesis of CM involves a complex interplay of genetic predispositions and environmental exposures, ultraviolet radiation exposure being the predominant environmental risk factor. The emergence of new biomarkers, such as novel immunohistochemical markers, gene mutation analysis, microRNA, and exosome protein expressions, holds promise for improved early detection, and prognostic and personalized therapeutic strategies.

## 1. Introduction

Cutaneous melanoma (CM) is a particularly aggressive form of cancer that originates from melanocytes, pigment-producing cells derived from the neural crest [1]. Despite representing a mere 4% of all skin cancers, CM accounts for up to 75% of skin cancer-related deaths [2]. However, with early detection and proper intervention, over 90% of the cases could be cured [3].

The pathogenesis of CM is multifactorial, involving both genetic and environmental factors [4]. Ultraviolet radiation (UVR), either from natural light or artificial sources, is the most important environmental risk factor for CM. Additionally, individuals with lighter complexion have the highest risk of developing CM due to lower levels of melanin which make these individuals more likely to develop sunburns. A higher number of nevi are also associated with an increased risk. A familial history of CM further increases this risk, possibly due to shared sun exposure behaviors or hereditary genetic mutations [5]. 

In this context, patient survival is strongly correlated with an early detection of the disease. Among various prognostic factors, the depth of invasion remains the most critical determinant of survival in numerous studies. Intraepidermal (in situ) melanomas can be cured by excision alone, and thin melanomas have minimal metastatic potential [6]. On the contrary, thick CMs still have very high mortality rates [7].

Despite its growing incidence, due to the advancements in diagnosis and management, the prognosis of CM has significantly increased. In this context, due to the ongoing challenges in melanoma prevention, diagnosis, and treatment, we performed a literature review discussing the latest progress in the diagnosis and management of CM, emphasizing the role of genetic testing, conventional immunohistochemistry, as well as several emerging biomarkers. These novel biomarkers such as microRNA and exosomes may significantly improve the prognosis of melanoma due to their potential to be used for non-invasive early diagnosis and monitoring, but also to become therapeutic targets.

## 2. Melanoma Pathogenesis

Ultraviolet radiation exposure is a major risk factor for CM due to the UVR capacity to damage DNA, causing somatic mutations [8]. The exposure can be classified as intermittent or chronic, the latter being mostly occupational. Both these exposure patterns are associated with an increased risk of CM, but it appears that the risk is higher for intermittent exposure [8,9]. This may be explained by the fact that intermittently exposed individuals have lower melanin levels and are more likely to develop sunburns [10]. Nevertheless, there are cutaneous melanomas, such as acral melanomas, which arise in skin that is not exposed to UVR. In this context, according to the 2023 WHO Classification of Tumors, cutaneous melanomas are classified as melanomas arising in sun-exposed skin and melanomas arising in sun-shielded sites (Table 1) [11].

Apart from UVR exposure, hereditary predisposition is another risk factor for cutaneous melanomas. However, familial cases encompass around 10% of all melanomas [12]. In this respect, several high-penetrance genes such as *CDKN2A*, *CDK4*, or *BAP1* are the most mutated in hereditary melanomas [13]. Individuals with germline mutations in the *CDKN2A*, a tumor suppressor gene, have a very high lifetime risk of developing CM, this mutation being encountered in up to 40% of melanoma-prone families [12]. Nevertheless, these mutations are relatively rare and are responsible for just around 2% of all CM cases [8]. In addition to these high-penetrance genes, some medium-penetrance genes such as *MITF* and *MC1R* are also involved in hereditary CM [12]. Furthermore, *MC1R* can be considered a “melanoma modifier gene” as it also increases the penetrance of *CDKN2A* [14].

As hereditary melanomas are relatively rare, most cutaneous melanomas are characterized by a remarkably high burden of somatic genetic mutations [15,16]. Identifying these genetic mutations can serve both diagnostic and prognostic purposes. Genetic testing is particularly useful for the diagnosis of dedifferentiated CMs which lack typical morphological and immunohistochemical features. In such cases, the diagnosis can be established by identifying melanoma-specific mutations [17].

The most frequent mutations in CM affect genes involved in the aberrant activation of the RAS/RAF/MEK/ERK signaling pathway, also known as the mitogen-activated protein kinase (MAPK) pathway, and the phosphoinositol-3-kinase (PI3K)/AKT pathway [18]. These mutated genes include *BRAF*, *NRAS*, *NF1*, *PTEN*, *KIT*, *TP53*, *CDKN2A*, and *TERT* [19,20]. 

The MAPK pathway is involved in the transduction of extracellular signals to the nucleus, thus activating genes that regulate cell proliferation and differentiation [21,22]. This aberrant activation is responsible for several cellular dysfunctions, such as the deregulation of the cell cycle and inhibition of apoptosis [21,23,24]. MAPK is the most frequently dysregulated pathway in cutaneous melanoma [25]. Up to 90% of all melanoma cases exhibit an abnormal activation of the MAPK pathway. The second most frequently activated pathway in CM is the PI3K pathway which plays a crucial role in maintaining cellular homeostasis [26,27].

As the MAPK pathway is the most affected in CM, numerous mechanisms contribute to its abnormal signaling, including *BRAF* mutations [18,28]. Between 37% and 60% of cutaneous melanomas harbor a somatic mutation in this gene, with the highest frequency observed in CM associated with low CSD [29]. The majority of *BRAF* mutations in cutaneous melanoma are missense, resulting in amino acid substitutions at the valine 600 position. Approximately 80% are V600E mutations (glutamic acid substitution), while 5–12% are V600K mutations (lysine substitution). Less common mutations include V600D (valine to aspartic acid) or V600R (arginine substitution). Additionally, *BRAF* non-V600 mutations can occur in around 5% of cases [30]. The *BRAF* gene encodes a protein kinase with three distinct domains: two regulatory and one catalytic. The latter is involved in the phosphorylation of MEK and in maintaining the protein inactive through a hydrophobic interaction. [31]. In the *BRAF* V600E mutation, the hydrophobic valine residue is substituted by a polar, hydrophilic glutamic acid which induces a conformational change in the catalytic domain, resulting in a constitutively active kinase [32,33]. *BRAF* non-V600E mutations generally operate through a similar mechanism, enhancing *BRAF* kinase activity [33]. Acknowledging these mutations is clinically significant for treatment and prognosis. *BRAF* V600-mutated melanomas can be treated with BRAF/MEK inhibitors, with response rates higher in V600E-mutated cases compared to V600K-mutated cases. Furthermore, even though the evidence is still limited, *BRAF* non-V600-mutated melanomas may still benefit from BRAF/MEK inhibitors [30].

The second most prevalent cause of aberrant MAPK pathway signaling In cutaneous melanoma is attributed to activating mutations in the *NRAS* gene. These mutations occur in 15–30% of melanomas and are predominantly missense, most often affecting codon 61 [34,35]. These mutations perpetuate aberrant signaling through both the MAPK and PI3K pathways [18,36,37]. It is noteworthy that *NRAS* and *BRAF* mutations are generally considered to be mutually exclusive, although co-mutations have been observed in rare instances [37]. *NRAS*- and *NRAS-BRAF*-co-mutated melanomas have a less favorable prognosis than *BRAF*-mutated ones as there are no target therapies for *NRAS* mutations [17].

Neurofibromin 1 (*NF1*) is a tumor suppressor gene, mutated in 10–15% of CM, making it the third most common mutation in this pathology [38,39]. *NF1* alterations are more frequent in melanomas associated with high CSD. These cases tend to possess a high mutational burden, including a co-occurrence of *BRAF* or *NRAS* mutations [19,40]. The NF1 protein serves as a regulator of the RAS family, attenuating downstream RAS signaling [41]. Consequently, loss-of-function mutations in *NF1* result in the hyperactivation of NRAS, leading to increased signaling through both the MAPK and PI3K pathways [19,38,39,41]. Analyzing *NF1* mutation status has some prognostic value even though there are no target therapies for *NF1*-mutated melanomas, but such cases respond favorably to immunotherapy [42]. Moreover, *NF1* analysis can offer important diagnostic information as this mutation is particularly common in dedifferentiated lesions which can be difficult to diagnose otherwise [17].

The receptor tyrosine kinase KIT plays a crucial physiological role in the proliferation and survival of melanoma cells, through the PI3K and the MAPK signaling cascades. *KIT* mutations are found in 2–8% of melanoma cases and are more common in acral melanomas and melanomas associated with low CSD [43,44]. Recognizing these mutations is important as such cases can benefit from tyrosine kinase inhibitors [45].

Mutations in the *TERT* promoter confer a proliferative advantage to melanoma cells and are common in advanced disease, being associated with a less favorable prognosis. Nevertheless, this mutation could become a potential therapeutic target [46,47]. *TP53*-mutated melanomas are also associated with advanced disease [46]. Assessing the status of *TP53* is important as these mutations have been associated with MAPK inhibitor resistance but they can also become potential therapeutic targets [48,49].

The *PTEN* gene, a tumor suppressor gene, is commonly dysregulated in the vertical growth phase of melanoma and in metastatic lesions, occurring in 10–30% of cutaneous melanomas [18,50]. *PTEN* alterations tend to be mutually exclusive with *NRAS* mutations but often co-occur with mutations in *BRAF* [51,52]. This co-occurrence has been hypothesized to increase PI3K pathway activation [51,52], mimicking the effects of an *NRAS*-only activation [51,53]. Additionally, *PTEN* loss-of-function is involved in acquired resistance to BRAF inhibitors in *BRAF*-mutated melanomas [54]. As mentioned before, *BRAF*-mutated melanomas may respond to BRAF inhibitors. However, therapeutic success is often temporary, as patients usually experience disease progression at some point or may even exhibit primary resistance to this target therapy. In this respect, acquired genetic mutations affecting the MAPK and PI3K signaling pathways play a central role in resistance to both chemotherapy and targeted therapies [55,56,57]. In this context, targeted *PTEN* therapy could improve the outcomes of the patients [49]. Having taken everything into consideration, due to this extraordinary genetic heterogeneity of melanomas, a multi-faced diagnostic and therapeutic approach including the identification of molecular biomarkers and genetic aberrations is imperative for optimizing patient outcomes.

## 3. Diagnostic and Prognostic Immunohistochemical Markers in CM

Cutaneous melanomas can manifest a broad array of morphological characteristics, rendering them easily confusable with other neoplastic lesions on standard histopathological examination. Consequently, additional diagnostic tools, particularly immunohistochemical (IHC) staining methods, may be necessary, especially in instances where the histological sample is partial, or the differentiation status of the neoplasm is ambiguous [58].

Various melanocytic markers such as S100, HMB45, Melan A, tyrosinase, MITF, and SOX10 can aid in the detection and subtyping of melanoma [58,59,60]. The S100 marker stands out for its high sensitivity for melanomas of all subtypes, including desmoplastic melanoma [60,61]. However, it is important to note that while S100 demonstrates high sensitivity, its specificity is limited, given that it is also expressed in a range of other malignancies and normal cellular components, such as dendritic cells, certain macrophages, and Schwann cells in lymph nodes [62,63]. This lack of specificity can create diagnostic pitfalls by masking the presence of small metastatic melanoma foci amid other S100-expressing structures within lymph nodes [62,63]. Additionally, primary cutaneous, particularly dedifferentiated ones, and metastatic melanomas can, in rare cases, lack S100 expression [17]. In this context, Aisner D.L. et al. discovered that approximately 1% of metastatic melanoma specimens were devoid of S100 expression. The loss of S100 expression did not appear to correlate with any specific histological subtype or the anatomical site of metastasis [64].

HMB-45 and Melan-A/MART-1 are melanocyte-specific markers with considerable specificity. HMB-45 recognizes gp100, a component of the melanosomal complex, and is highly specific for melanoma [65,66,67]. HMB-45 is particularly useful in distinguishing between a benign and a malignant melanocytic tumor as nevi gradually lose HMB-45 due to their maturation process [67,68]. Nevertheless, its use is limited due to low sensitivity as it fails to stain a significant proportion of metastatic melanomas [69,70]. These numbers can be remarkably high even in primary cutaneous melanomas with divergent differentiation, when HMB-45 may be negative in up to 86% of the cases [17]. Melan-A expression is predominantly localized within the endoplasmic reticulum and melanosomes, thus having higher sensitivity compared to HMB-45. Melan-A is particularly useful in identifying isolated tumoral melanocytes in the dermis, which can reclassify a melanoma initially diagnosed as in situ to an invasive lesion [66,71,72]. Drabeni M. et al. reported an increased Breslow thickness in approximately 60% of cases when utilizing Melan-A compared to H&E staining alone [71]. Similarly, Megahed M. et al. found evidence of dermal invasion in 30 out of 104 cases that were initially classified as melanoma in situ based on H&E staining [72]. However, Melan-A analysis has its limitations. The formation of pseudomelanocytic nests—clusters of Melan-A positive cells at the dermo–epidermal junction—can confound the diagnosis of melanoma in situ in the presence of lichenoid inflammation [73]. The concomitant application of nuclear markers like MITF (microphthalmia-associated transcription factor) and SOX10 has been suggested as a solution [73]. SOX10 is significantly more specific than Melan-A (96% vs. 17%) in identifying epidermal melanocytes and consequently avoiding the overdiagnosis of melanoma in situ in sun-damaged skin [74]. Similarly, MITF is also superior to Melan-A for the correct diagnosis of solar lentigo [75]. Additionally, Melan-A also fails to stain most primary dedifferentiated cutaneous melanomas. SOX10 and MITF perform better in such cases but their sensitivity is still rather low at around 34% [17]. In this context, melanomas completely lacking expression of conventional melanocytic markers have been described in both primary and metastatic lesions [76]. In addition to their low sensitivity in dedifferentiated lesions, none of these markers is entirely specific for melanomas [17]. For example, clear-cell sarcomas express HMB-45, Melan-A, MITF, S100, and SOX10 [77,78]. PEComas, even though rarely located on the skin, express HMB-45 and MITF [79]. Malignant peripheral nerve sheet tumors express SOX10 and S100, and in rare cases can express Melan-A and tyrosinase [80]. As a consequence, several other immunohistochemical markers have been developed.

The NK1/C3 antibody is noteworthy for its ability to identify a specific cytoplasmic antigen prevalent in melanoma cells. The NK1/C3 antibody was synthesized at the Netherlands Cancer Institute, and although its target antigen was not initially known, it appears to be a glycoprotein located on the membranes of cytoplasmatic vesicles in melanoma cells [81,82]. However, its expression is not confined to melanoma alone. It is also detected in other melanocytic lesions including intradermal and compound nevi, congenital nevi, dysplastic nevi, blue nevi, and Spitz nevi [81]. Moreover, NK1/C3 is also highly sensitive for metastatic lesions [70]. Nevertheless, this antigen is also sporadically present in certain non-melanocytic neoplasms such as a subset of breast and prostate carcinomas, cellular neurothekeomas, granular cell tumors, and dendritic cells within lymph nodes [81,83,84]. Furthermore, the high cost associated with its use poses an additional impediment to its widespread use [85]. These constraints underscore the need for further research to either refine the specificity of NK1/C3 or to identify alternative, cost-effective markers with greater diagnostic precision. Future studies might aim to delineate the functional implications of the antigen identified by NK1/C3 in tumor pathogenesis, as this could provide additional insight into its potential roles as a therapeutic target or as part of a multi-marker diagnostic panel. 

Immunohistochemical analysis for PRAME (preferentially expressed antigen in melanoma) has become increasingly used in the diagnosis of CM [86,87,88]. One study assessed the immunoexpression of PRAME in 400 melanocytic tumors, including primary melanomas, metastatic melanomas, and melanocytic nevi [86]. The study revealed diffuse nuclear immunoreactivity in over 80% of the metastatic (87%) and primary melanomas (83.2%). The expression was notably high in all subtypes except for desmoplastic melanomas (35%). Importantly, PRAME was expressed in both in situ and non-desmoplastic invasive components. Additionally, 86.4% of the melanocytic nevi investigated were entirely negative for this marker [86]. Furthermore, PRAME analysis is also useful for identifying melanocytic pseudonests in order to distinguish melanoma in situ from lichenoid dermatoses or keratoses [89]. PRAME expression also seems retained in primary dedifferentiated melanomas, but more studies are needed to confirm these findings [17]. Moreover, it is significantly less expressed in clear-cell sarcoma, PEComas, and other skin spindle-cell neoplasms that can be considered in the differential diagnosis of CM [90,91]. Therefore, PRAME is a constituent of a 23-gene array diagnostic assay used for cutaneous melanoma [92,93], and is one of two genes employed in a noninvasive molecular assay that aids clinicians in determining the necessity for biopsy in the case of melanocytic lesions [94]. PRAME is also highly useful for assessing metastatic lesions. For instance, a study comparing nodal nevi and melanoma metastases demonstrated that PRAME was expressed in 0% of the nevi and in 100% of the lymph node metastases [87]. Nevertheless, its expression is not confined to melanoma but is also present in various other malignancies, such as lung cancer, breast carcinoma and other gynecological malignancies, renal carcinoma, leukemia, synovial sarcoma, myxoid liposarcoma, and various other sarcomas [95,96,97,98]. Therefore, in the setting of metastatic disease, the diagnosis of melanoma should not solely rely on PRAME expression and should be confirmed by additional immunohistochemical or molecular analysis. However, PRAME analysis for metastatic melanoma may not possess just diagnostic, but also prognostic value as it may become a potential target for immunotherapy [99].

In addition to these markers, BRAF V600E immunohistochemical analysis is increasingly performed for primary cutaneous melanomas and metastatic lesions [100,101]. This method is particularly useful for diagnosing dedifferentiated melanomas that lack expression of conventional melanocytic markers and may aberrantly express other markers [102]. Another great advantage of immunohistochemical analysis is its considerably lower costs compared to molecular analysis while the results are similar, with a study reporting a sensitivity of 80.8% and specificity of 100% for immunohistochemistry [103]. In spite of these promising results, at present BRAF V600E immunohistochemical results should be confirmed by PCR analysis, as concordance between the two methods varies between 71.4 and 97% [101]. In the future, immunohistochemical methods for BRAF detection may significantly improve, providing increased diagnostic accuracy at lower costs.

The main characteristics of the immunohistochemical markers discussed above are presented in Table 2.

Another pitfall in the immunohistochemical analysis of CM is that rare cases may display atypical IHC staining patterns. As mentioned before, these may include the aberrant expression of markers typically unrelated to melanocytes or the absence of conventional melanocytic markers, further complicating the diagnosis [104]. The heterogenous immunophenotypic profile in melanoma underscores the necessity for a multi-marker approach to the diagnosis of this malignancy. It also raises important questions about the biological mechanisms underlying the loss of conventional melanocytic markers during disease progression. These aspects could have significant implications for both prognosis and therapeutic choices. Future research might focus on understanding the molecular mechanism of marker loss, helping the development of more effective diagnostic tools and targeted therapies.

Finally, apart from their diagnostic value, immunohistochemical markers have also been analyzed as prognostic tools. Even though traditional markers may lose expression in dedifferentiated and metastatic lesions, it is not clear how this correlates with prognosis. For instance, primary dedifferentiated cutaneous melanomas and conventional CMs with similar prognostic factors have a similar overall prognosis [105]. In recent years, as PRAME expression has been associated with prognosis in patients with uveal melanomas [106], it has been studied as a potential prognostic marker in CM but it does not influence patient survival [107,108]. On the contrary, a higher expression of the proliferation marker, Ki67 has been associated with decreased survival in some studies [107,109]. A 2021 meta-analysis found a significant association between higher Ki67 expression and lower overall survival rates. However, no correlation was found between Ki67 expression and progression-free survival or recurrence-free survival [110] Having taken all these reports into consideration, Ki67 may be proved to be a useful prognostic factor in CM, but further studies are needed to validate these findings and establish a cut-off value associated with decreased survival. 

## 4. Emerging Biomarkers in CM

Due to the extraordinarily heterogenous histopathological, immunohistochemical, and molecular landscape of CM, this disease continues to pose important challenges in terms of diagnosis and treatment. Therefore, several new biomarkers have become increasingly studied in recent years in the hope of improving the understanding of CM pathogenesis and management. Unlike immunohistochemical and genetic testing, these emerging biomarkers are expected to improve the early detection and subsequent monitoring of CM in rapid, cost-effective, and non-invasive ways as they can easily be analyzed from blood samples. Furthermore, these new biomarkers may also serve as potential therapeutic targets.

### 4.1. MicroRNA

MicroRNA (miRNAs) represent non-coding RNAs involved in degrading mRNAs [96]. They have been increasingly recognized as critical modulators of oncogenic processes, including various stages of cancer progression such as melanoma [111,112]. In melanomas, miRNA dysregulation is involved in promoting cell proliferation, resistance to apoptosis and invasion, angiogenesis, and metastasis [113]. Additionally, miRNAs have also been associated with resistance to BRAF and MAPK inhibitors [114,115]. In this context, due to their detectability in both intra- and extracellular compartments and their stable levels even in unfavorable conditions, miRNAs have attracted considerable attention as emerging biomarkers in oncology [116]. 

At present, miRNA levels can be assessed from various sources, such as resected primary or metastatic tumors, as well as arterial or venous plasma and serum. Importantly, the data derived from these different sources have shown no significant divergence. Depending on the type of cancer under investigation, abnormal miRNA expression profiles have been found to correlate with various disease stages, overall prognosis, tumor recurrence, and potential responsiveness to therapeutic interventions [111,117]. In patients with melanoma, different miRNAs can be either up- or down-regulated, and have been correlated with progression-free survival and overall survival [118,119,120]. Interestingly, serum levels of various microRNAs can discriminate between melanoma stages with increased accuracy compared to S100B or LDH [121]. For instance, miR-137, miR-148, and miR-182 downregulate MITF expression and promote tumor invasion [122]. miR-221 plasma levels are increased in melanoma patients, and are correlated with stage, recurrence, and disease progression [123]. Rigg E. et al. demonstrated that miR-146a-5p is overexpressed in melanoma brain metastases and its knockdown results in a reduction of metastatic lesions [124]. On the contrary, other types of microRNA such as mirR-211, miR-542 3p, or miR-152-3p are downregulated in invasive melanomas [125]. In vitro studies demonstrated that increasing miR-152-3p expression inhibits the proliferation and invasiveness of melanoma cells [126]. miR-542 3p is involved in epithelial-to-mesenchymal transition (ETM), and its experimental upregulation inhibited ETM and metastatic spread [127]. miR-143 also bears anti-tumoral effects as it has been linked to promoting apoptosis and inhibiting the proliferation of melanoma cells [128]. Similar anti-tumoral effects have been reported for miR-224-5p which can additionally overturn acquired resistance to BRAF inhibitors [129]. Several other microRNA seem to play a role in resistance to target or conventional chemotherapy. A downregulation of miR-7, miR-579 3p, and miR-126 3p was found in melanomas resistant to BRAF/MAPK inhibitors [130,131,132], while miR-31 downregulation is associated with chemoresistance [133]. Importantly, the experimental upregulation of miR-7 and miR-126 3p restored responses to BRAF inhibitors in melanoma cell lines [130,131].

Therefore, miRNA analysis could become a useful tool for monitoring melanoma progression after surgical excision and therapy, as well as a potential therapeutic target.

### 4.2. Exosomes

Exosomes are extracellular vesicles secreted by cells, encompassing a unique molecular signature that reflects the cell type from which they originate. Given their traceable cellular origins and facile isolation, exosomes can be regarded as potential biomarkers for diagnosis and prognosis in various cancers, including melanoma [134]. They are readily available through minimally invasive methods, as they can be isolated from a variety of biological fluids such as blood, plasma, urine, and cerebrospinal fluid [135]. Exosomes extracted from melanoma cell lines have been shown to contain distinct mRNA, miRNA, and protein profiles [134,135]. Exosome analysis can offer important diagnostic and prognostic information, as various exosomal components are significantly altered in cutaneous melanomas [134,136,137]. In this respect, Surman M. et al. found increased exosome concentrations in melanoma cases but those levels were not correlated with disease stage [134]. On the contrary, Boussadia Z. et al. reported a higher exosome concentration in metastatic melanoma compared to non-metastatic cases [138]. The complex relationship between exosomal components and melanoma progression is not entirely understood, but various mechanisms have been proposed. For instance, exosomes can carry and modulate the activity of matrix metalloproteinases (MMPs), as well as alter cell adhesion and activate fibroblasts to become cancer-associated fibroblasts, thus stimulating melanoma invasiveness [134,139,140,141]. Exosomal components have also been shown to enhance metastatic potential in CM by promoting epithelial-to-mesenchymal transition (ETM) [142], angiogenesis [134,143], and lymphangiogenesis [144]. As melanoma is particularly prone to brain metastases, exosomes may also at least partially explain this characteristic by damaging endothelial cells and the blood–brain barrier, and activating glial cells [145]. 

Some exosomal components have also been linked to resistance to therapy in CM, and targeting these molecules may improve therapeutic response [146]. In this context, exosomes can influence the melanoma microenvironment by altering the function of lymphocytes and stimulating tumor-associated macrophages (TAMs) to become M2-polarized and secrete pro-tumorigenic cytokines [147,148,149]. These effects can affect the response to immunotherapy, and targeting TAMs could improve the outcome of melanoma patients [150]. Furthermore, exosomes can be used as a means for administering therapy [151]. In this respect, exosomes containing BRAF siRNA were shown to have increased anti-tumoral activity compared to siBRAF in melanoma cell lines [152]. Similarly, cord-blood-derived exosomes produced significant genotoxicity and a decrease in survival time for melanoma cells and lymphocytes from melanoma patients, apparently by delivering anti-oncogenic miR-7. These results are particularly important as the exosome caused no significant damage to normal lymphocytes [153]. While these findings underscore the promising role of exosomes as diagnostic, prognostic, and treatment tools in melanoma, additional research is required to comprehensively delineate their utility. Future studies may aim to validate these biomarkers in larger patient cohorts, completely elucidate the roles of exosomal components in melanoma progression, and assess the feasibility of incorporating exosomal markers into existing diagnostic, prognostic, and therapeutic frameworks.

The complex effects of exosomal components in CM pathogenesis are presented in Figure 1.

### 4.3. Melanoma-Inhibiting Activity

Melanoma-inhibiting activity (MIA) is a soluble protein overexpressed in melanoma cells and actively secreted into the extracellular environment, where it binds to various extracellular and cell surface proteins [154]. This protein was first identified in supernatants of melanoma cell lines and, in vitro, it was considered to possess growth-inhibiting activities [155]. Despite its paradoxical nomenclature, in vivo, elevated levels of MIA have been substantiated to promote invasive capabilities, extravasation, and metastatic spread [156]. Recent studies on murine melanocytes demonstrated that MIA is involved in cellular senescence, and its knockdown enhances cell proliferation [157].

In a study encompassing 176 cutaneous melanoma patients, a progressive escalation in serum MIA levels was observed in correlation with advanced stages of the disease. Only 18.5% of patients in stage I displayed elevated MIA levels, as opposed to 59% in stage IV [156]. Alegre E. et al. found significantly increased MIA levels in patients with metastatic CM compared to disease-free patients or healthy individuals [158]. Similar results were reported by various other authors [159,160,161]. Furthermore, increased MIA levels are significantly associated with decreased survival [158,162]. Moreover, MIA levels have also been correlated with melanoma recurrence [163]. Compared to LDH, measuring MIA concentrations is a more accurate method for identifying patients with advanced disease and for predicting metastatic spread in CM [160]. In this context, as MIA concentrations are correlated with disease severity, this protein may become a prognostic biomarker in cutaneous melanomas. Furthermore, MIA could also become a therapeutic target itself, as Schmidt J. et al. demonstrated that inhibiting MIA dimerization resulted in the reduction of melanoma metastases in murine models [154].

It is, however, crucial to recognize that serum MIA levels are not exclusively elevated in melanoma. Elevations have also been documented in lung cancer [164], while immunohistochemical analysis demonstrated positive MIA expression in lung, esophageal, and cervical cancers [159]. 

## 5. Conclusions

Cutaneous malignant melanoma is a prevalent and highly aggressive form of skin cancer that requires improved prevention, diagnosis, and treatment methods. Recognizing risk factors such as UVR exposure, genetics, and family history is crucial for prevention. 

The role of traditional biomarkers remains essential in the diagnosis and monitoring of melanoma. However, the field is rapidly evolving with the identification of emerging biomarkers. MicroRNA, exosomes, and MIA offer insights into melanoma pathogenesis and progression, potentially serving as both diagnostic and therapeutic targets. These emerging biomarkers could be the key to more personalized and effective treatments, ultimately improving survival rates and quality of life for patients.

## Figures and Tables

**Figure 1 ijms-24-15881-f001:**
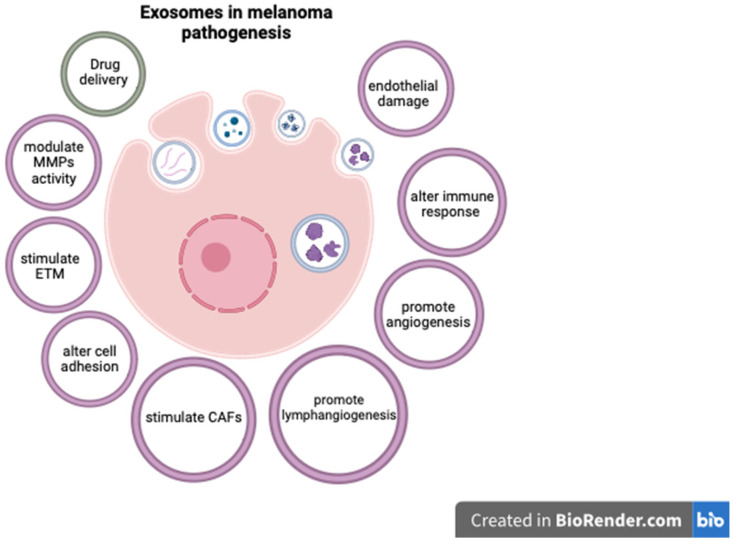
The role of exosomal components in CM (created with Biorender, https://www.biorender.com, accessed on 29 October 2023).

**Table 1 ijms-24-15881-t001:** WHO Classification of cutaneous melanomas.

Melanomas arising in sun-exposed skin	Low CSD melanoma: SSM, low CSD nodular melanoma
	High CSD melanoma: lentigo malignant melanoma, high CSD nodular melanoma
	Desmoplastic melanoma: most often associated with severely sun-damaged skin
Melanomas arising in sun-shielded skin or without known UVR exposure	Spitz melanoma
	Acral melanoma
	Melanoma arising in congenital nevus
	Melanoma arising in blue nevus

CSD—cumulative sun-damage, SSM—superficial spreading melanoma.

**Table 2 ijms-24-15881-t002:** Advantages and disadvantages of the main melanocytic markers.

Melanocytic Marker	Advantages	Disadvantages
S100	High sensitivity for all CM melanoma subtypes (including desmoplastic) and for metastases	Limited specificity: expressed in normal cells in the lymph nodes (dendritic cells, macrophages) and non-melanocytic tumors
HMB-45	High specificity for CM and for melanoma metastasis to lymph nodes	Limited sensitivity: fails to stain primary dedifferentiated CM and metastatic melanomas; Expressed in some non-melanocytic tumors: clear-cell sarcoma, PEComa
Melan A	Higher sensitivity compared to HMB-45; Improves Breslow depth evaluation	Can stain pseudomelanocytic nests, resulting in a false-positive diagnosis; Limited sensitivity in primary dedifferentiated CM; Expressed in clear-cell sarcoma, MPNST
MITF	Higher sensitivity and specificity than S100 and HMB-45	Relatively low sensitivity for dedifferentiated melanomas; Expressed in clear-cell sarcoma, PEComa
SOX 10	Generally high sensitivity and specificity	Relatively low sensitivity for dedifferentiated melanomas; Expressed in clear-cell sarcoma, MPNST
NK1/C3	High sensitivity	Low specificity; Expensive to use
PRAME	High sensitivity for primary and metastatic melanomas, including dedifferentiated lesions	Low sensitivity for desmoplastic melanomas; Relatively low specificity, particularly for metastatic lesions
BRAF V600E	Useful for dedifferentiated melanomas; Lower costs than PCR analysis	Sensitivity and specificity need to be improved

## Data Availability

Not applicable.

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
