# Peer review of "Cutaneous Melanoma: A Review of Multifactorial Pathogenesis, Immunohistochemistry, and Emerging Biomarkers for Early Detection and Management"

_ijms, 2023, doi:10.3390/ijms242115881_

Round 1
Reviewer 1 Report
Comments and Suggestions for Authors
Smart review. This is a good, concise, updated and clear review about the different melanocytic biomarkers, considering sensitivity and specificity. Well written and well balanced. This is a contribution.
Minor points to be addressed that do not need a second-round review.
Line 57: Replace Melaomas by melanomas.
Inside Table: maliga by malgnant. I am not sure about the use of “maligna” in English.
Line 67: families [12]. (Point is missing
Line 71. Delete the point and repair the citation marks in the expression considered a melanoma modifier gene.
Line 88: Up to 90% of melanoma cases….. Please, indicate if that is referred to any type of CM, somatic or familiar CM.
Line 209: deddiferentiated by dediferentiated.
Line 217: Is there any extra information about the nature of the antigen recognized by NK1/C3 antibody.
Table 2: PRAME: Hlgh sensitivity …. By High sensitivity for primary.
Line 343: MIA. Please, justify the name, as it is stated thast the amount of this protein is higher in advanced melanoma.
Line 352: recurrence by reccurence.
Comments on the Quality of English Language
Some editing needed
Author Response
Summary. We want to thank the reviewer for their observations. The revised paper contains changes according to comments. As indicated, we highlighted what was added to make it easier to track our changes. In addition, the text was subjected again to grammar and spelling checking – all modifications are highlighted as indicated.
Comments 1. Smart review. This is a good, concise, updated and clear review about the different melanocytic biomarkers, considering sensitivity and specificity. Well written and well balanced. This is a contribution.
Minor points to be addressed that do not need a second-round review.Line 57: Replace Melaomas by melanomas.Inside Table: maliga by malgnant. I am not sure about the use of “maligna” in English.
Line 67: families [12]. (Point is missing
Line 71. Delete the point and repair the citation marks in the expression considered a melanoma modifier gene.
Line 88: Up to 90% of melanoma cases….. Please, indicate if that is referred to any type of CM, somatic or familiar CM.
Line 209: deddiferentiated by dediferentiated.
Table 2: PRAME: Hlgh sensitivity …. By High sensitivity for primary.
Line 352: recurrence by reccurence.
Response 1. Thank you for your observations. We have made the required changes and revised the quality of the English language with the help of professional translator.
Comments 2. Line 217: Is there any extra information about the nature of the antigen recognized by NK1/C3 antibody.
Response 2. Thank you for your suggestion. We have provided additional information about the NK1/C3 antibody and the antigen it recognizes.
Comments 3. Line 343: MIA. Please, justify the name, as it is stated thast the amount of this protein is higher in advanced melanoma.
Response 3. Thank you for your recommendation. We have added more information explaining the history of this protein’s name.
Reviewer 2 Report
Comments and Suggestions for Authors
A review by Gosman et al presents a current status of knowledge on melanoma pathogenesis and detection, including most prominent biomarkers. Overall, the manuscript is written well, however, the Authors should underscore more clearly differences between more recent discoveries and those, which are known and established considering for example melanoma detection.
Specific comments:
1. Gene names should be consistently written in italics.
2. Paragraph 4 is important: (1) it is unknown why the Authors focus on these particular features. (2) both miR and exosomes should be more broadly discussed with regard to melanoma.
3. The manuscript would benefit from preparing any figure.
Comments on the Quality of English Languageminor revision required
Author Response
Summary
We want to thank the reviewer for their observations. The revised paper contains changes according to comments. As indicated, we highlighted what was added to make it easier to track our changes. In addition, the text was subjected again to grammar and spelling checking – all modifications are highlighted as indicated.
Comments 1. A review by Gosman et al presents a current status of knowledge on melanoma pathogenesis and detection, including most prominent biomarkers. Overall, the manuscript is written well, however, the Authors should underscore more clearly differences between more recent discoveries and those, which are known and established considering for example melanoma detection.
Response 1. Thank you for your report. In response to your thoughtful feedback, we have carefully emphasized the differences and the utility of using emerging biomarkers in comparison to conventional ones. Therefore, we have expanded the Introduction section as well as section 4. Emerging biomarkers in CM.
Comments 2. Gene names should be consistently written in italics.
Response 2. Thank you for your observation. We have made the required changes.
Comments 3. Paragraph 4 is important: (1) it is unknown why the Authors focus on these particular features. (2) both miR and exosomes should be more broadly discussed with regard to melanoma.
Response 3. Thank you for your suggestions. In consideration of that, we have significantly expanded section 4. Emerging biomarkers in CM to include more relevant references and to highlight the importance of focusing on miR and exosomes as both of them could significantly improve CM management.
Comments 4. The manuscript would benefit from preparing any figure.
Response 4. Thank you for tour recommendation. In response to your previous suggestions of extending section 4. Emerging biomarkers in CM, we have added a figure highlighting the complex interplay between exosomes and CM.